# Evaluation of Stem Rust Disease in Wheat Fields by Drone Hyperspectral Imaging

**DOI:** 10.3390/s23084154

**Published:** 2023-04-21

**Authors:** Jaafar Abdulridha, An Min, Matthew N. Rouse, Shahryar Kianian, Volkan Isler, Ce Yang

**Affiliations:** 1Bioproducts and Biosystems Engineering Department, University of Minnesota, 1390 Eckles Ave, St. Paul, MN 55108, USA; 2U.S. Department of Agriculture, Agricultural Research Service, Cereal Disease Lab, 1551 Lindig Avenue, St. Paul, MN 55108, USA; 3Department of Computer Science, University of Minnesota, 100 Union St SE, Minneapolis, MN 55455, USA

**Keywords:** hyperspectral camera, wavelength, vegetation indices, classification, reflectance

## Abstract

Detecting plant disease severity could help growers and researchers study how the disease impacts cereal crops to make timely decisions. Advanced technology is needed to protect cereals that feed the increasing population using fewer chemicals; this may lead to reduced labor usage and cost in the field. Accurate detection of wheat stem rust, an emerging threat to wheat production, could inform growers to make management decisions and assist plant breeders in making line selections. A hyperspectral camera mounted on an unmanned aerial vehicle (UAV) was utilized in this study to evaluate the severity of wheat stem rust disease in a disease trial containing 960 plots. Quadratic discriminant analysis (QDA) and random forest classifier (RFC), decision tree classification, and support vector machine (SVM) were applied to select the wavelengths and spectral vegetation indices (SVIs). The trial plots were divided into four levels based on ground truth disease severities: class 0 (healthy, severity 0), class 1 (mildly diseased, severity 1–15), class 2 (moderately diseased, severity 16–34), and class 3 (severely diseased, highest severity observed). The RFC method achieved the highest overall classification accuracy (85%). For the spectral vegetation indices (SVIs), the highest classification rate was recorded by RFC, and the accuracy was 76%. The Green NDVI (GNDVI), Photochemical Reflectance Index (PRI), Red-Edge Vegetation Stress Index (RVS1), and Chlorophyll Green (Chl green) were selected from 14 SVIs. In addition, binary classification of mildly diseased vs. non-diseased was also conducted using the classifiers and achieved 88% classification accuracy. This highlighted that hyperspectral imaging was sensitive enough to discriminate between low levels of stem rust disease vs. no disease. The results of this study demonstrated that drone hyperspectral imaging can discriminate stem rust disease levels so that breeders can select disease-resistant varieties more efficiently. The detection of low disease severity capability of drone hyperspectral imaging can help farmers identify early disease outbreaks and enable more timely management of their fields. Based on this study, it is also possible to build a new inexpensive multispectral sensor to diagnose wheat stem rust disease accurately.

## 1. Introduction

Cereal crops are suffering from multi-diseases that are spreading worldwide. Rusts are an essential disease in wheat crops caused by the fungal genus (*Puccinia graminis* f. sp. *Tritici*); it can attack most cereal crops, not only wheat crops, and is also called black rust. The infection occurs through wind and humidity, with some other optimal environmental conditions. Stem rust disease can progress quickly in crops to cause serious yield losses. The variety of rust classes involved with cultural practices, cultivar resistance, meteorological conditions, and a changing pathogen population results in a significant variation in the severity of yearly rust epidemics [1]. Economic losses can be severe (generally up to 70%) due to the damage of the photosynthesis part on diseased leaves, resulting in reduced grain production and increased shriveling, which influence grain quality and quantity. In extreme conditions of an actual susceptible cultivar attached with early contaminations and conducive climatic conditions, stem rust can result in 70% crop loss [2]. Therefore, determining plant disease status is necessary to understand how a particular disease situation affects the crop. Accurate field disease evaluations could improve yield by informing management decisions, such as fungicide applications, and research decisions, such as which lines to advance in breeding. Some parameters should be considered to evaluate the incidence and severity of the disease. Disease incidence is the proportion of unhealthy plants with the total number of plants investigated. It is formulated in terms of percentage. In contrast, disease severity is the proportion of infected plant tissue, also measured as a percentage. Disease severity is also described as infection intensity. In a few situations, the evaluation of disease severity, such as vascular wilts of annuals, cereal smut, neck blast, and brown rot of stone fruits, could be calculated directly based on crop yield loss [3]. In the field, the severity of the disease varies based on different plant pathosystems, and pathosystem-specific indices and ratings have been used. Often, percentage scales and standard area diagrams of disease intensity are used to guide disease ratings. Plant pathologists generally rely on the Weber–Fechner law, which states that visual activity depends on the logarithm of the intensity of the stimulus. However, the human eye has limitations, and the naked eye is inaccurate in distinguishing diseased plants [4]. The observation by human eyes is inaccurate and can cover only 50% of appearance symptoms compared to other methods; therefore, we always need an assistance method to evaluate the disease severity. A 12-grade scale has been suggested [5] for all diseases of plants. As mentioned above, visual abilities are limited to distinguishing disease severity throughout a huge area. The human scouting method is a common method to detect the disease, but it needs to use trained people and the hiring of more expert people in a small area. Since human scouting is limited, remote sensing is now preferred due to labor cost, efficiency, and repeatability for detecting diseases in growers’ fields. Remote sensing can also be an effective tool for complementing or possibly replacing disease scouting.

Remote sensing is a fast, non-destructive, and cost-effective procedure that can obtain and evaluate the properties of ground surfaces from different distances. For techniques using spectral imaging, spectral data can be acquired from satellites, airplanes/UAVs, or land-based systems. The results of previous studies [6] have demonstrated a promising impact on agriculture by improving crop protection. When plants experience stressful conditions, ranging from drought, infection, nutrient deficiency, etc., the spectral reflectance captured by remote sensing provides unique information to identify the stress types. Several techniques have been studied to provide more information about disease detection, such as visible, infrared, multispectral and hyperspectral imaging, multi-band and fluorescence spectroscopy, nuclear magnetic resonance spectroscopy, fluorescence imaging, thermography, etc. Remote sensing technologies will significantly spatialize diagnostic results and thus ensure that agriculture practices are more sustainable and environmentally friendly by preventing excessive use of costly pesticides in crop protection.

Spectral analysis has been an essential method for disease detection in terms of remote sensing. Zhang et al. [7] studied an emerging spectral analysis method—continuous wavelet analysis—and showed that the wavelet features could capture significant major spectral signatures of rust disease. Bohnenkamp et al. [8] detected and measured the quantity of stripe rust disease in winter wheat using a UAV in the field. Using real-time data from field measurements can guide plant protection measures and enhance the use of resources. In a different study, Bebronne et al. [9] considered three common diseases in wheat crops. The diseases were stripe (rust), brown (leaf) rust, and septoria tritici blotch (STB). The reflectance was measured by a multispectral camera, and partial least squares regression (PLSR) and artificial neural networks (ANN) were performed to classify the severity of the diseases. STB disease obtained the highest classification value among other diseases [10]. Krishna et al. [11] utilized hyperspectral remote sensing to study winter wheat in multistage disease infestations. They used a field spectroradiometer over the spectral range from 350 to 2500 nm. Two classification methods were employed: partial least squares (PLS) and multiple linear regression techniques (MLR) to distinguish suitable wavebands and create spectral models for measuring the severity of stripe rust disease in winter wheat. Guo et al. [12] agreed that hyperspectral imaging technology has proven to be successful in detecting plant diseases. Vegetation indices are the most common indicator that can be used in the remote sensing technique; thus, hundreds of bands could be reduced and or removed from the data set. Vegetation indices have been used to clarify the most important wavelength that can detect stresses on plants, and many studies have approved vegetation indices are valid for this purpose. Zheng et al. [13] determined the most sensitive bands for stripe rust disease detection in multiple wheat growth stages, such as 460–720 nm in the early-mid growth stage, and 568–709 nm and 725–1000 nm in the mid–late growth stage. Two vegetation indices PRI and ARI (Anthocyanin Reflectance Index) were performed at different growth stages and evaluated to decide whether they could be used for estimating the severity of wheat stripe rust disease. Huang et al. [14] evaluated the accuracy of the photochemical reflectance index (PRI) to quantify the disease index (DI) of stripe wheat rust in wheat and its applicability in the detection of the disease using hyperspectral imagery.

Ashourloo et al. [15] utilized vegetation indices to evaluate stripe rust in winter wheat, using a digital camera to extract fractions of the various disease symptoms. Wenjiang et al. [16] applied vegetation indices that were measured during growth duration and found that stripe rust disease had a logarithmic relationship between the ratios of Transformed Chlorophyll Absorption in Reflectance Index (TCARI) to Optimized Soil Adjusted Vegetation Index (OSAVI). The photochemical reflectance index (PRI) was used as an indicator of stripe rust disease in winter wheat. Several studies showed that hyperspectral sensors are valuable tools for disease detection, identification and quantification at different scales, from the tissue to the canopy level [17].

Since most studies have focused on stripe stem rust, stem rust disease has been investigated in this study for infected and non-infected wheat plants. The study evaluates the difference in spectral reflectance among four disease classes which corresponded to inoculated and fungicide-protected wheat plots and different wheat genotypes with varying resistance responses to stem rust (healthy, mildly diseased, and severely diseased). Detection of stem rust is a critical approach in informing mitigation strategies. The main purpose of this study is to develop a drone-image processing technique for detecting stem rust disease levels, including low disease severities. This study will help plant breeders and pathologists to identify stem rust disease severity levels in trial plots using aerial images instead of, or to complement, using manual labor.

The study objectives are to: (i) utilize hyperspectral images to classify wheat stem rust disease levels (healthy, mildly diseased, and severely diseased), and (ii) select the best wavebands and vegetation indices to determine the disease responses by using a hyperspectral camera mounted on a UAV. The result of hyperspectral imaging was compared to the ground truth based on traditional measurements of stem rust disease.

## 2. Material and Methods

### 2.1. Field Experimental Design

In 2020, plot trials were planted at the University of Minnesota Rosemount Research Outreach Center in Rosemount, MN, USA, with 6 blocks each containing 150 plots, each representing a unique wheat line (Figure 1a,b). Three of the six blocks were inoculated with the stem rust pathogen (*Puccinia graminis* f. sp. *tritici*) race TPMKC (isolate 74MN1409), and the rest were treated with a fungicide spray regime to prevent the wheat plants from infection and provide fully healthy plants. Fungicide Tilt (Syngenta) was applied at the stem elongation growth stage. Fungicide Folicur (Bayer) was applied at the heading stage). Within each block, four sub-blocks were divided by spreader row of cultivars “Morocco” and ”Baart”, and each sub-block contained 40 four-row trial plots. The size of each trial plot was 1 m × 1 m. In each block, 150 wheat genotypes were included that exhibited various stem rust disease resistance levels from very resistant to very susceptible. Thus, each wheat genotype was included in each of the six blocks. The wheat genotypes included 120 inbred lines derived from a cross between spring wheat cultivars “Linkert” and “Forefront” in addition to several hard red spring wheat cultivars adapted to Minnesota. Inoculations were executed as described previously [18] by spraying a light mineral oil suspension of urediniospores onto the spreader rows of the three blocks that were not treated with fungicides. Visual estimates of disease severity and infection response were recorded three times throughout the end of the growing season (while lines were in between watery–ripe and hard dough growth stages) as previously described [19]. Coefficients of infection were determined by calculating the product of the disease severity values and the linearized infection response values according to [20]. Disease severity was assessed on the 0 to 100% modified Cobb scale, whereas infection responses were classified into categories of resistant, moderately resistant, moderately susceptible, or susceptible based on pustule size and amount of chlorosis [21]. When multiple infection responses were observed in the same plot, all infection responses were recorded, with the most frequent infection response listed first.

### 2.2. The Features of Hyperspectral Camera and UAV

Spectral image data were collected in the field by a UAV (Matrices 600 Pro, Hexacopter, DJI Inc., Shenzhen, China) (Figure 2) and a hyperspectral imaging system (Resonon, Inc., Bozeman, MT, USA); the flight computer (Resonon NUMI, flight code version 1.36) was attached to the camera system had a 33° field of view. The images were uploaded from the flight computer to the desktop by using a cable. Resonon Pika ll g visible and near-infrared camera and spectral range (400–900 nm) is a “push-broom” or line-scan type imager that produces a 2-D image with spectral resolution 2.1; the spatial channel is 640 and spectral channel 240, where every pixel in the image contains a continuous reflectance spectrum. The HS camera also included an inertial measurement unit (IMU; Ellipse2-N, SBG Systems, Carrières-sur-Seine, France), a single band GNSS (Global Navigation Satellite System) antenna (TW2410, Tallysman, Inc., Ottawa, ON, Canada). The camera’s orientation was controlled by a three-axis gimbal (Ronin-MX, DJI, Inc., Shenzhen, China). Ground elevation was maintained before flying by a level scale to ensure the camera was in the correct position. The UAV and HS camera were powered by individual lithium polymer batteries, that were used for each piece of equipment in the system.

### 2.3. UAV Image Acquisition

Preflight steps were conducted to get high-resolution images. A KMIL file was created for the area of interest. KMIL was uploaded directly from the Resonon software to the camera. There were a few steps that need to be carried out in the Resonon Ground Station software (version 3.123): (i) the camera must be in the right position; (ii) camera performance must be checked before applying data collection in the field, so manual recording should be conducted; and (iii) the gimbal should keep the heading of the hyperspectral imager relative to the heading of the UAV so that the spectral image array (640 spatial channels) could remain approximately perpendicular to the heading of the UAV flight direction. The gimbal should also maintain the elevation angle of the camera to avoid distorted images. The camera would only start recording when the drone imaging system was in the target area created in the initial KIML file, which was a polygon created by google earth.

### 2.4. Flight Plan

Autonomous flight tasks were conducted using DJI Ground Station Pro (iOS app version 1.8.2 + for 2018 flights and iOS app version 2.0.2 + for 2019 flights) software. The altitude was set at 20 m, ground speed at 4 m/s, ground swath at 8 m, pixel size at 3 cm, and cropped plot size at 0.914 m. Flights were only conducted in good weather, with minimal cloud and low-speed wind. Twelve reference panels made of high-density fiberboard (60 cm × 60 cm × 3.2 mm) were distributed immediately before spectral image acquisition, the reference panels were strategically placed throughout the experimental area so that images containing reference panels were captured at least every 90 s to account for temporal variation in solar illumination.

### 2.5. Image Preprocessing

Spectronon software (Version 2.134; Resonon, Inc., Bozeman, MT, USA) was utilized to correct raw images to radiance by using the radiance conversion plugin. Following radiometric correction, images were georectified (using the georectification plugin) based on time-synced data collected by the GNSS receiver and IMU of the airborne system (i.e., latitude, longitude, altitude, yaw, pitch, and roll). A 1.0 m digital elevation model image was used as the basis for projecting each image line to the appropriate elevation above mean sea level. Then, the georectified radiance images were converted to reflectance by applying a single spectrum correction to each image based on the relationship between the radiance of the reference panels from the imagery and the average measured reflectance. Radiance pixels were manually extracted to avoid pixels near the edges of panels. If there was no reference panel present in an image, radiance data extracted from the reference panel captured closest in time were used for reflectance conversion.

### 2.6. Image Post-Processing and Spectral Extraction

A total of 40 images were captured in this study and were stitched together. Spectral extraction was accrued manually for each plot. Plots were divided into 4 categories: healthy (class 0), mildly diseased (class 1), moderately diseased (class 2), and severely diseased (class 3), based on the ground-truth coefficient of infection values averaged from multiple ratings. The sample size for classes 0–3 was 700 for each class, and the sample size was the spectral reflectance measurement. When training classification models, the larger the sample size, the more accurate the classification result. Each extracted spectrum contained 30–35 pixels. The Savitzky–Golay smoothing procedure (11-band moving window, second-order polynomial fitting) was applied to remove noise in the spectra.

### 2.7. Classification and Feature Selection Methods

In this study, five classification methods were applied to confirm that the results were correct. Furthermore, the methods were utilized previously in several studies in agriculture and remote sensing. Four classes were processed through machine learning (Healthy, class 1, 2, and 3). The algorithms were run 10 times, and their performances were assessed based on the average classification accuracies (overall and individual class). The classification methods were performed as follows: 70% training and 30% testing; the sample sizes were between 699 and 784 spectral data samples. The classification methods were: quadratic discriminant analysis (QDA), random forest classifier (RFC), support vector machine (SVM), and decision tree classification (DTC). Several researchers have applied QDA in remote sensing and spectral reflectance (e.g., [22,23]). The QDA is a quadratic function model; thus, the covariance matrices for each stem rust disease level were estimated using the training data’s maximum likelihood. Then algorithms of QDA and RFC were practical to the covariance matrices in order to find the most excellent discriminant rate between healthy and unhealthy plants. QDA and RFC are flexible in nature and can be used for both classification and regression techniques. Compared to other machine learning techniques like SVM, Gaussian Naïve Bayes, DTC, and linear discriminant analysis, RFC had better accuracy with fewer image data [24].

There were 240 bands in the dataset; therefore, we applied feature selection to select the best bands to clarify the classification accuracy with the best band choice to detect the stem rust disease at both high and low severity levels, using the Scikit-learn machine learning developed in the Python package [25,26].

Feature selection is the biggest challenge in choosing the most effective bands in order to determine the most sensitive band to detect stem rust disease. Feature selection should be determined based on which classification method can distinguish between healthy and non-healthy plants. We used Scikit-learn, version 0.9.1, to provide a different feature selection method and, most importantly, relative features. Several steps were applied to get the best bands selection, such as the Chi-Square Statistic and ANOVA F-value, to obtain feature importance metrics. Different methods selected a variety of most important features and related feature lists with the importance decent sequence. Based on each method, a feature list was provided. We used QDA, RFS, DTC, and SVM algorithms with K-fold cross-validation and used mean accuracy scores to evaluate the best band combination. Best bands selection was performed for the highest classification method.

### 2.8. Testing Wavelength

Two parameters were utilized to explore in which wavelength the significant differences occurred in order to distinguish diseases and disease development stages: (i) sensitivity value, which was calculated as the mean reflectance value of diseased leaves divided by the mean reflectance value of healthy leaves (at each wavelength); (ii) linear correlation coefficient (r), which visualizes the intensity of each spectrum band.

We also applied several classification methods and the variation between vegetation indices to distinguish the differences between healthy and non-healthy plants. The sensitivity values and correlation coefficient were supportive methods that distinguished between healthy and non-healthy plants.

### 2.9. Spectral Vegetation Indices (SVIs)

Since the hyperspectral camera has numerous bands, scholars have developed and created several SVIs for quality and quantity evaluation of vegetative covers. Vegetation indices can minimize the complex mixture of vegetation, soil, background, shadow, plastic cover reflection, and brightness, especially when it has a white plastic cover, soil reflection, and soil color. In the last decade, scientists have developed some SVIs to identify disease in several stages since some disease symptoms have unique fluctuations in band reflection. For example, some symptoms affect the pigments or chlorophyll content, which will affect the green band reflection, affecting the bands in the 500–600 nm range. The SVI can be calculated by ratioing, differencing, ratioing differences and sums, and by finding linear groupings of spectral band data. Based on the calculation of each SVI, it can give a value indicating the plant’s health status. There were 14 vegetation indices applied and calculated to select the best one that could detect stem rust disease (Table 1). The classification methods were applied to identify the disease coefficient of infection in multiple classes. The procedure of selecting the best bands was the same as for the VIs selection method.

#### Variance

The 4 wheat stem rust disease classes can produce some variance in the spectra among the classes based on disease development or plant verities.

Variations were calculated for specific disease severity levels by Equation (1).
(1)$$Variance\;\left(x\right)=\frac{1}{N}\sum_{i=1}^{N}{\left(xi-mi\right)}^{2}$$
where *x* is a single variate, *m* is the known mean value, and *N* is the number of data collected.

The calculation of SVIs was based on a few wavelengths, and the impact of the disease progression for each SVI was estimated. The method to find the SVI relies on a few wavelengths to evaluate the drivers’ plant parameters. The effect of the disease on SVIs was evaluated to understand each SVI’s response to disease severity. The variation of the 14 SVIs was calculated individually as a standard to consider SVIs at different disease severity.

## 3. Results and Discussion

### 3.1. Wavelength in Visible and Near-Infrared Range for Binary Classes (Healthy vs. Infected Plants)

The spectral reflectance obtained from the hyperspectral images can be divided into two ranges, visible (400–690 nm) and near-infrared (NIR), 691–900 nm. In this study, Figure 3a shows that the visible range in green and red ranges is obviously higher than in healthy plants. So, it was clear that the spectral reflectance increased in the green and red range, and this distortion continued to the red edge at 710 nm. Plots in class 1 showed green leaves during hyperspectral measurement with few symptoms of wheat stem rust disease. The near-infrared range had a different scenario, so the spectral reflectance of the diseased plant was high in the green and red ranges, while it was low in NIR. First, the differences started from the red edge to 800 nm. The spectral reflectance of healthy plants recorded the highest reflectance value of about 25%, while stem rust disease was recorded at a lower value of 20%. In this case, a plot with more symptoms would mainly impact the spectral reflectance in the NIR range.

In reality, many wheat plots showed severe symptoms, and this impacted spectral reflectance. In general, the maximum and minimum of some spectral values are shown in the curve, which indicates that spectral reflectance changed based on plant symptoms. This process would be beneficial for breeders to evaluate and monitor the disease levels of different crop varieties. The plant breeder needs to plant different wheat varieties to select the best variety for future cultivation. In addition, early disease detection can be helpful in reducing and/or making fungicide applications more efficient during growing seasons [38].

### 3.2. Wavelength in Visible and Near-Infrared Range for Multiclass

The spectral reflectance curve shows the spectral reflectance behavior between healthy and other spectral reflectance data under stress. The spectral reflectance can be divided into two ranges, visible (400–690 nm) and near-infrared (NIR) 700–900 nm (Figure 3b). In this study, Figure 3a,b shows the fluctuation of the visible range was not obvious between healthy and class 1 and 2, but for class 3. the situation was different, so it was clear the spectral reflectance increased in the green and red range, and this distortion continued to the red edge (710 nm). The severity of disease in class 1 and 2 showed green leaves during hyperspectral measurement with few symptoms of wheat stem rust disease. The coefficient infection of class 1 was ranked between a 1–15 score, so it was very difficult to see any symptoms in this stage. The near-infrared range had different scenarios based on disease severity; therefore, the peak of each severity stage was more variable than other classes. First, the differences started from the red edge to 800 nm. The spectral reflectance of healthy plants recorded the highest reflectance value of about 25%, while class 3 was recorded as a lower value of 20% spectral reflectance. In this case, when the disease has more symptoms, they impact the spectral reflectance in the NIR range.

### 3.3. Disease Sensitivity

#### 3.3.1. The Sensitivity

Disease sensitivity or disease ratio was utilized to test the wavelength activities (Figure 4a). Class 3 recorded the maximum value was 700 nm and the curve clearly showed unique pattern for each stage. Class 2 showed lower values than healthy and class 3. In Figure 4a, the most differences were in visible green range (530 nm) and red edge (690 nm). The red edge 690–730 nm was a critical range for high disease severity, and the peak of the disease ratio reached up to 1.2. Disease sensitivity helped to show obvious variation between the three classes and healthy, so the ratio value could be used to distinguish between several varieties in visible range might not shows clearly in spectral reflectance curve.

#### 3.3.2. Linear Correlation Coefficient

The correlation coefficient of class 1 and class 2 was almost identical until 700 nm, then the correlation started to change and drop down, but they looked to have the same pattern (Figure 4b). Class 3 showed a different pattern than other classes, so it had a lower correlation value in all bands, especially in 735 nm. The linear correlation coefficient also supported the previous result and showed wide differences between class 3 and two other classes. In reality, the wheat plants showed severe symptoms in class 3, which impacted the spectral reflectance. In general, in remote sensing, the maximum and minimum of some spectral values were shown in some curves and gave a good indicator that spectral reflectance had been changed based on plant symptoms. This process can be very helpful for breeders to evaluate and monitor plants in different varieties, such as tolerant, mild, and susceptible. If this technique is applied in a field application, it will be an important scientific breakthrough.

### 3.4. Classification Accuracy and Feature Selection

#### Multiclass Classification Results

The highest classification accuracies of healthy and several classes were achieved by random forest classifier (RFC) and decision tree classification (DTC). Both methods recorded the highest accuracy value in class 2, 97% and 95% for FRC and DTC, respectively. Quadratic discriminant snalysis (QDA) and support vector machine (SVM) achieved lower accuracies than RFC and DTC in class 3 and other classes (Figure 5). The most critical disease level that remote sensing helps growers and breeders in field inspection was class 1 (mildly diseased), in which the disease existed, but the symptoms were not visible to human eyes. However, since class 1 is not that different from class 0 (healthy), the classification accuracies for RFC and DTC were relatively low (76% and 69%, respectively) (Figure 5). Our results show that classification of low disease levels is feasible, indicating that early detection of stem rust is possible. Previous studies of hyperspectral images have detected laurel wilt disease even in asymptomatic stages [39]. Remote sensing is one of the precision agricultural keys to timely identifying rust development, mitigating its yield effects, and applying remedial action in a site-specific manner [40].

Detecting diseases in their early stages will prevent them from spreading widely throughout the region. Therefore, binary classification was performed in addition to the disease severity classification to detect the diseased plants from healthy plants. RFC and DTC achieved the highest classification accuracy among all classifier methods. The classification rates of RFC were 76%, 81%, 84%, 97%, and 88% (Figure 5). Since the RFC achieved the highest classification rate, feature selection from RFC was used to select wavebands. In the binary classification, the best bands in feature selection were distributed in the different ranges of NIR, red, red edge, and blue (817.5, 612.2, 760, 725.1, 497.2, and 706.6) (Table 2). As mentioned in the Section 2, wavelengths were selected upon the high rate accuracy of any classification methods. In this study, the highest classification method was RFC, compared to other classification methods. The best bands of multiclass are displayed for research purposes since we have multiple varieties of wheat crops with such tolerance, susceptibility, etc. The order of each wavelength selected was placed based on the weight in the classification order, starting from 85 to 100%. The values inside the parentheses represent the weight value of the wavelengths. In other words, when the band has a higher value, such as 99%, it means that it has the highest classification value to distinguish the healthy from non-healthy. Hyperspectral cameras also showed the capability to discriminate wheat rust infection using machine learning algorithms such as SVM and, thus, represent a possibility for wheat rust early detection [41].

Different diseases may have other impacts on plants. For example, some diseases affect the water content or chlorophyll content, and this will affect the spectral reflectance [42] in chlorophyll bands (green range—550 nm) or water absorption in 900 nm [43]. Frank et al. [44] focused on a few relevant bands, and the detection accuracy was enhanced. Therefore, more reliable information could be extracted, which may be helpful in agricultural practices. In addition, by applying band selection techniques, fungal infection of different varieties and healthy wheat stands could be accurately differentiated from infected areas [45]. Similarly, feature selection increases the classification accuracy and computational efficiency in processing and analyzing hyperspectral images [46]. Therefore, minimizing hundreds of bands could develop a new sensor with multispectral or multi bands (6 bands or less) for several agricultural purposes. Designing a filter depends on feature selection to distinguish the best bands applicable to identify the disease or stress type and level. Park et al. [47] have applied feature selection to minimize redundancy and maximize relevance (mRMR) for feature selection techniques to directly choose essential raw bands from hyperspectral images (777.24 nm, 547.77 nm, 474.32 nm, 859.45 nm, and 735.85 nm).

A hyperspectral camera is still an expensive tool in agriculture, so it is necessary to build an inexpensive remote sensor. As mentioned above, there are several varieties, and each stage has different symptoms based on the morphological phenotyping of the plant.

### 3.5. Vegetation Indices

#### Classification Accuracy and Best SVIs Selection

Fourteen SVIs were analyzed to identify which SVI is suitable for classifying diseases associated with different physiological plant parameters. The classification accuracy varied between classes, in this case, RFC achieved higher classification rates than other classification methods for all disease classes and overall classification (Figure 6). The highest overall classification accuracy was 76% for RFC. Therefore, the best vegetation indices were selected based on RFC among all classifiers.

In this study, the most significant SVIs were GNDVI and RARSc (Table 3). Green-normalized differential vegetation index and ratio analysis of reflectance spectra (RARSc) accurately distinguished healthy plants from other classes, so GNDVI and RARSc have been noted as the best VSI and always top-ranked when we repeated the calculation process. Chl green came behind RARSc as the best VSI. NPQI also appeared as an important SVI in classifying all disease levels.

Franke et al. [48] indicated that NDVI and chlorophyll indices could distinguish the rust infection in individual wheat leaves spots based on associated changes in photosynthetically active biomass. Detecting the wheat stem rust disease by SVIs could help scientists to target only a few SVIs. However, RARSc and Chl green were chosen for disease severity classification, which means that chlorophyll content was impacted by the increasing rust lesions. Therefore, the infected wheat plants gradually lost chlorophyll content and prematurely turned yellowish and brownish. Many studies have mentioned that chlorophyll content is the best SVI to detect stress in wheat plants. In this study, there were several SVIs driven from green bands such as RARSc, Chl green, NPQI, RVSI, and PRI. Wheat stem rust disease also affected the concentration of carotenoids and the photoprotective role of xanthophyll pigments [49] related to low sunlight usage efficiencies [50]. When the disease is in an advanced stage, the pigment of the leaves and stems starts to change to a brown and dark brown color. Stem rust, known as black rust, reduces the yield and quality of seeds. Changing color affects the spectral reflectance wavelength range from 550 to 650 nm. The PRI is sensitive to variations in carotenoid colors (e.g., xanthophyll pigments) in live vegetation. Carotenoid color concentration is indicative of photosynthetic light use efficiency or the rate of carbon dioxide uptake by foliage per unit energy absorbed by chlorophyll. Therefore, any change in chlorophyll concentration will cause a difference in the value of PRI [51]. Due to this sensitivity of the PRI, it could be used to distinguish a diseased plant from a healthy plant and identify clear differences between them. Furthermore, the PRI represents changes in the xanthophyll cycle and can detect changes in yellow or brown carotenoid plant pigments that cause the chlorosis of leaf color [52]. The Red-Edge Stress Vegetation Index (RVSI) was also mentioned as sensitive VI to stem rust disease.

The red-edge vegetation stress index (RVSI) might quantify the stress on plants and describe the inflection point of the chlorophyll red edge [33]. Wójtowicz et al. [53] created random forest models that were verified by separately utilizing spectral reflectance. Three vegetation indices, namely CRI, PRI, and GNDVI, appeared to be the most useful for distinguishing uredinia from other symptoms on rye and wheat leaves.

### 3.6. Variance of Vegetation Indices

According to Figure 7, all SVIs showed different variance values between healthy and advanced stages of diseases. None of the vegetation indices have the same variance for healthy and other classes, which means that when the disease advanced, the spectral bands were affected. For example, some vegetation indices of healthy plants showed higher or lower values than plants with high disease severity. The most significant differences were between healthy plants and class 3 (severely diseased). Even with mild symptoms, obviously, there is variation between different classes of experiment plots. Identifying disease progress is a very complex process and would benefit from daily observation to evaluate plant status. Therefore, remote sensing would be the best way to help the breeders to evaluate field plots with reduced labor and with higher accuracy to identify the disease severity. Similarly, remote sensing can aid growers in identifying diseases and informing mitigation strategies. Minimizing the number of bands would be a great approach in remote sensing to prepare multi-band sensors to detect and distinguish plant stress levels.

## 4. Conclusions

A non-destructive method was applied to evaluate stem rust disease severity in different genotypes of wheat. Ground truth was compared with spectral reflectance bands and indices. Four classification methods were conducted in this experiment. For the wavelength extraction, the RFC classifier achieved higher accuracy than other classifiers. The classification of RFC recorded higher classification than other classification methods for multi-classes and binary classification. This classification rate is also considered an acceptable rate even for mild symptoms. The best six bands in binary classification were selected to distinguish healthy and none healthy plants (817.5, 612.2, 760, 725.1, 497.2, and 706.6). Spectral vegetation indices were calculated and classified. The most frequent SVIs were RARSc, GNDVI, PRI, RVSI, NDVI, and Chl green. These SVIs recorded the highest classification rate, and they were recorded in most classes and binary classifications. Reduction of data dimensions is necessary to focus on a few bands with less redundancy and noise. It is essential to more carefully consider the ability of SVIs for the detection of plant diseases, specifically for the diseases with diverse symptoms and for discrimination among different diseases. The critical approach for this study is the ability to detect stem rust disease in very low levels of the disease and very low coefficient infection scores (1–15). Therefore, the results are promising for precision agriculture to help growers and scientists to estimate the disease presence in the field.

This paper developed a drone hyperspectral imaging method for discriminating stem rust disease levels, including low disease severities. The models built in this study will help plant breeders and pathologists to identify the stem rust disease severity levels in trial plots using aerial images instead of or in complementation to using manual labor. Farmers could benefit from this study by taking timely actions based on early disease detection capabilities, as demonstrated by the detection of low disease severities. Future work based on feature band selection can construct a filter that can distinguish wheat stem rust disease from other biotic and abiotic stresses. This technique could also help to create a map for the infestation area to set up a plan for chemical application (fungal or pesticide applications). Early detection and timely decisions will reduce the negative impact of stem rust disease. However, the accurate and real-time early detection of wheat rust infection is still challenging before the crop shows associated symptoms.

## Figures and Tables

**Figure 1 sensors-23-04154-f001:**
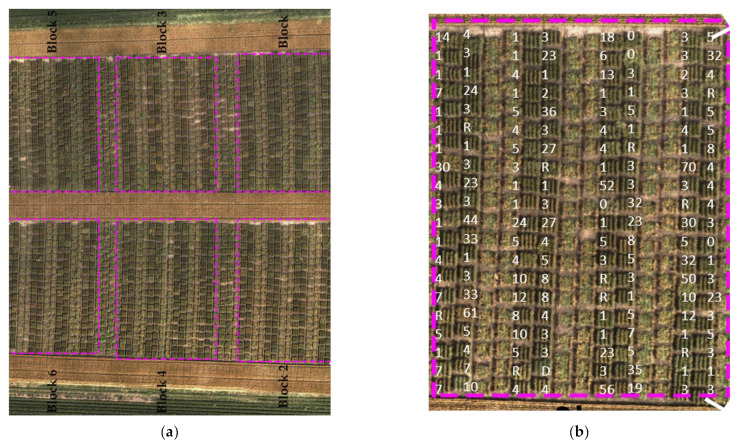
Top view of the field of the study in Rosemount, Minnesota: (**a**) aerial image of the entire experiment, blocks 2, 3, and 6 were inoculated, and the other blocks 1, 4, and 5 were sprayed with fungicide; (**b**) one *P. graminis*-inoculated block with a superimposed coefficient of infection values.

**Figure 2 sensors-23-04154-f002:**
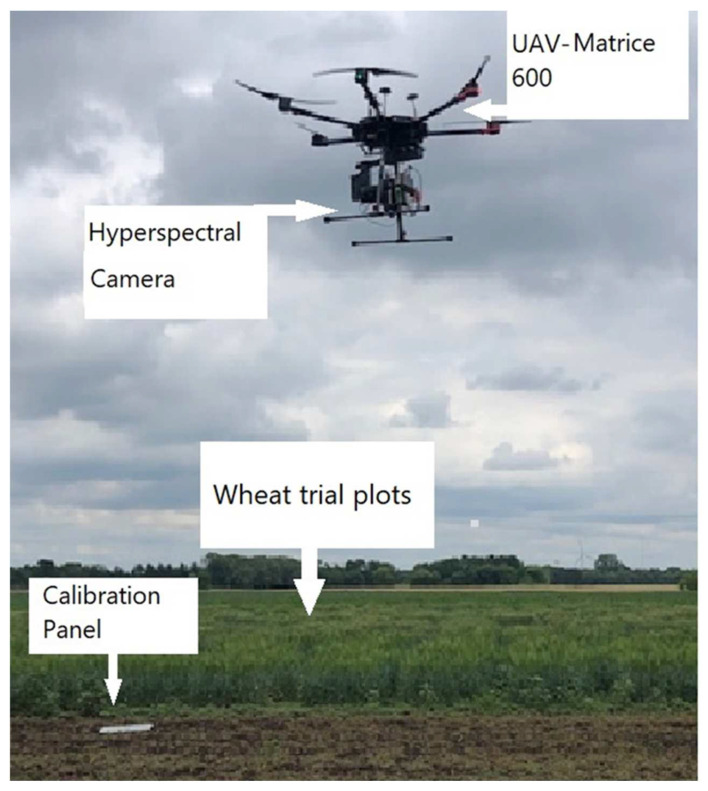
Unmanned aerial vehicle in the wheat field and calibration panels.

**Figure 3 sensors-23-04154-f003:**
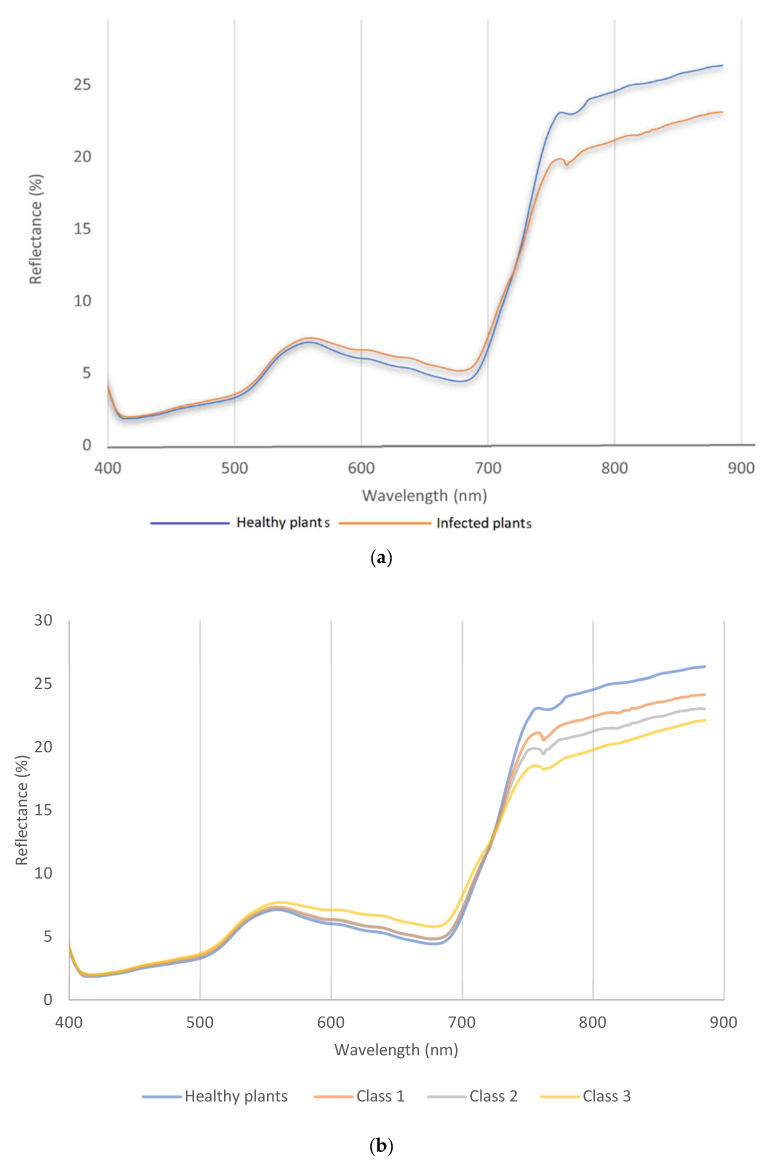
The spectral reflectance curve for wavelength vs. reflectance of wheat plants in stem-rust-inoculated blocks and wheat plants in the fungicide-treated blocks (healthy plants) for (**a**) binary classification and (**b**) multiclass (class 1, 2, and 3).

**Figure 4 sensors-23-04154-f004:**
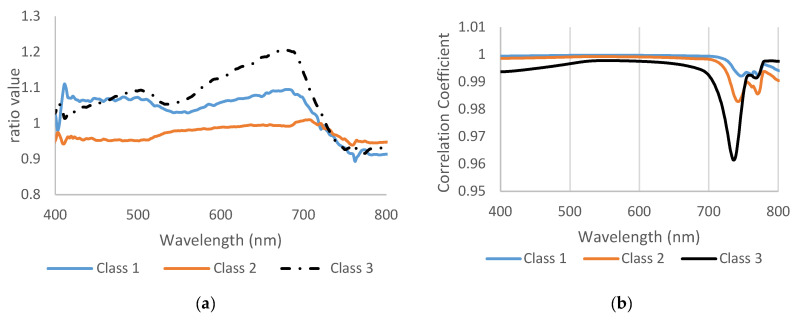
(**a**) The wavelength ratio or disease sensitivity and (**b**) the correlation coefficient between healthy and infected plants.

**Figure 5 sensors-23-04154-f005:**
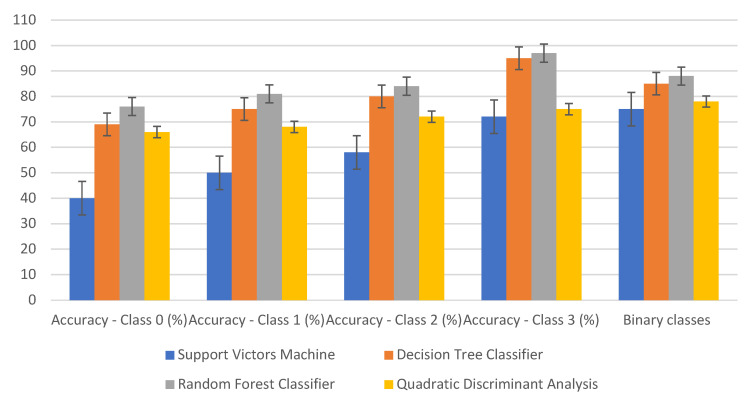
The results of multiclass classification and binary class with error bars among classes using classifiers SVM, DTC, RFC, and QDA. Class 0 is the healthy class, and classes 1–3 are mildly, moderately, and severely diseased classes.

**Figure 6 sensors-23-04154-f006:**
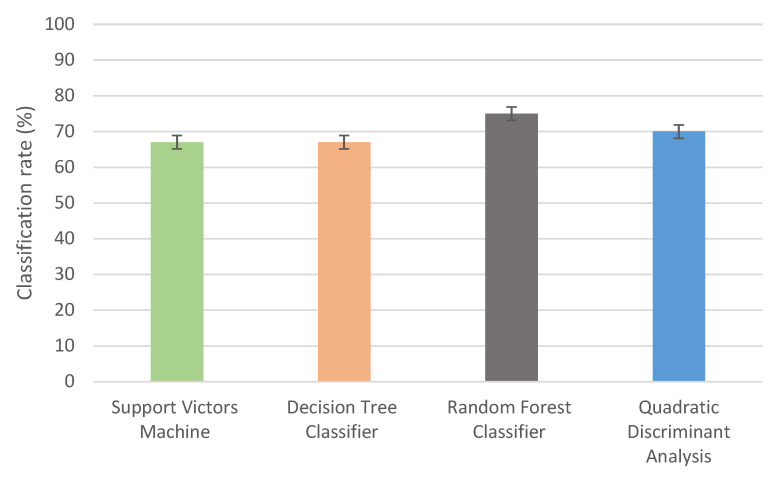
The overall classification accuracy of vegetation indices by applying different classification methods: SVM, DTC, QDA, and RFC.

**Figure 7 sensors-23-04154-f007:**
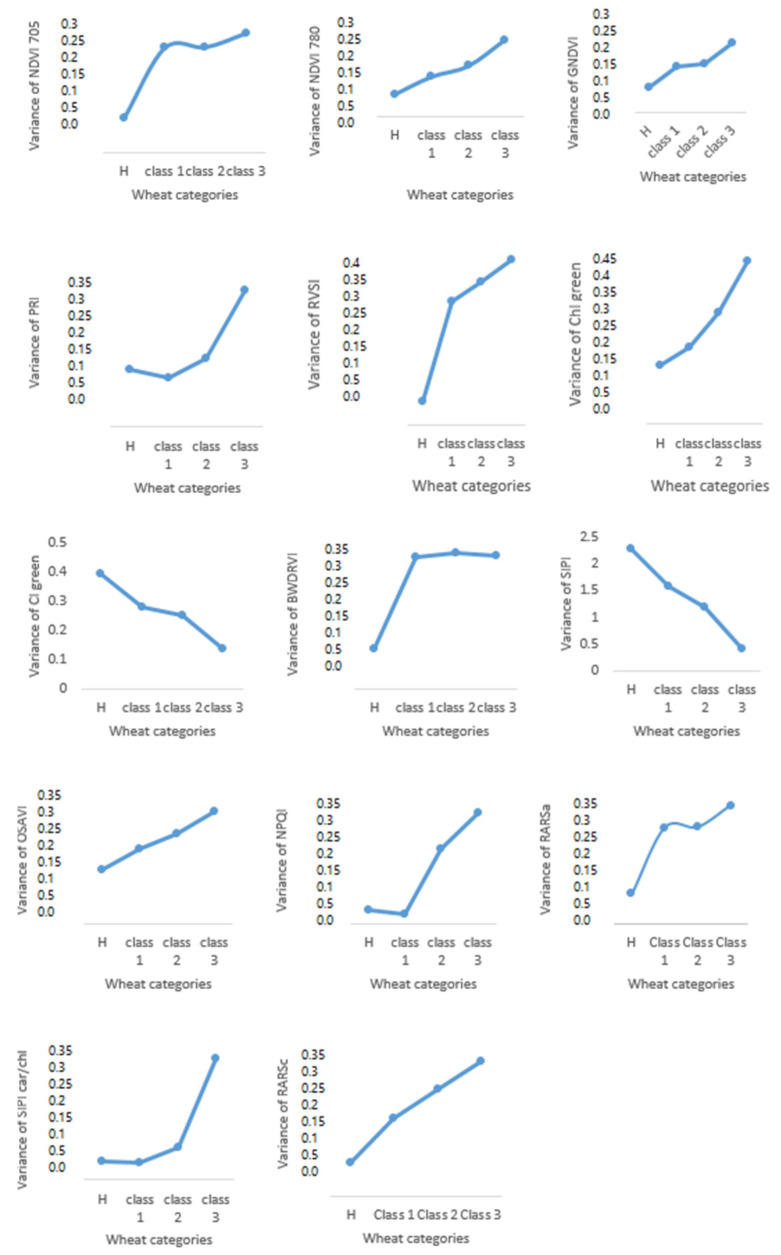
The variation of fourteen vegetation indices in three different classes.

**Table 1 sensors-23-04154-t001:** Spectral vegetation indices, equation, and resources.

*Vegetation Indices*	*Equations*	*References*
** *Ratio Analysis of reflectance Spectral Chlorophyll-a (RARSa)* **	$RARSa=\frac{R675}{R700}$	[27]
** *Ratio analysis of reflectance spectra (RARSc)* **	$RARSc=\frac{R760}{R500}$	[27]
** *Pigment-specific simple ratio (PSSRa)* **	$PSSRa=\frac{R800}{R680}$	[28]
** *Normalized difference vegetation index 780 (NDVI 780)* **	$NDVI\;780=\frac{R780-R670}{R780+R670}$	[29]
** *Green NDVI (GNDVI)* **	$GNDVI=\frac{\left(NIR850-R580\right)}{\left(NIR850+R580\right)}$	[30]
** *Structure Insensitive Pigment Index (SIPI)* **	$SIPI=\frac{\left(R840-R450\right)}{\left(R840-R670\right)}$	[31]
** *Normalized phaeophytinization index (NPQI)* **	$NPQI=\frac{\left(R415-R435\right)}{\left(R415+R435\right)}$	[32]
** *Normalized Difference Vegetation Index (NDVI 780)* **	$NDVI780=\frac{\left(R780-R671\right)}{\left(R780+R671\right)}$	[29]
** *Red-Edge Vegetation Stress Index (RVS1)* **	$RVSI=\left[\frac{\left(R714+R\;750\right)}{2}\right]-R\;733$	[33]
** *Chlorophyll Green (Chl green)* **	$Chl\;green=(\frac{\left(R780\right)}{\left(R540\right)\;}$)^−1^	[34]
***Chlorophyll Index Red-Edge (CIrededge*710*)***	$CI\;rededge710=\left(\frac{R780}{R710}\right)-1$	[34]
** *Blue-wide dynamic range vegetation index* **	*BWDRVI =* $\frac{\left(0.1\;*\;800-Blue\right)}{\left(0.1\;*\;800+Blue\right)}$	[35]
** *Optimized Soil Adjusted Vegetation Index* **	*OSAVI = ((1 + 0.16) × ((R800 − R670))/(R800 − R670 + 0.16))*	[36]
** *Photochemical Reflectance Index (PRI)* **	$PRI=\frac{\left(R531-R570\right)}{\left(R531+R570\right)}$	[37]

**Table 2 sensors-23-04154-t002:** Best wavelength selections were chosen in different severity classes and binary classes using random forest classifier (RFC).

Categories	Best Bands Selection (nm) and the Weight of Each Band Inside the Parentheses
Class 1 (score 1–15)	811.3 (99%), 690.2 (97%), 764.1 (95), 774.4 (94%), 739.5 (90%), 682 (90%)
Class 2 (score 16–34)	762.1 (100%), 749.7 (98%), 404.9 (96%), 421.3 (93%), 409 (90%), 415.1 (89%)
Class 3 (score 35–70)	860.6 (99%), 714.8 (97%), 776.4 (93%), 710.7(91%), 760 (90%), 782.6 (87%)
Multi-classes (score 1–70)	817.5 (99%), 612.2 (97%), 760 (96%), 725.1 (94%), 497.2 (91%), 706.6 (90%)

**Table 3 sensors-23-04154-t003:** The result of VIs selection in random forest classifier for multi-classes and binary classification.

Categories	Best Vegetation Indices and the Weight Values Inside the Parentheses
Class 1 (score 1–15)	GNDVI (100%), PRI (98%), RARSc (97%), Chl green (93%), SIPI (90%), RVSI (88%)
Class 2 (score 16–34)	RVSI (98%), Chl green (96%), GNDVI (93%), NDVI 705 (91%), SIPI, RARSa (89%)
Class 3 (score 35–70)	GNDVI (100%), CI rededge 710 (97%), RVSI (95%), OSAVI (92%), PSSRa (92%), NDVI 780 (88%)
Binary classes	GNDVI (100%), PRI (98%), Chl green (96%), NPQI (94%), RVSI (90%), BWDRVI (88%)

## Data Availability

Data available on request due to restrictions eg privacy or ethical.

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
