# Peer review of "Evaluation of Stem Rust Disease in Wheat Fields by Drone Hyperspectral Imaging"

_sensors, 2023, doi:10.3390/s23084154_

Round 1

Reviewer 1 Report

Very good work!! Congrats!
I have a few comments

Line 18 and line 311 write "quadratic data analysis (QDA)," but the references (Sankaran and Ehsani 2013; Khaled et al. 2022) and the entire literature call it "quadratic discriminant analysis (QDA)." In fact, you also use "Quadratic Discriminant Analysis" in Figures 5 and 6 and in line 486.

Line 431 put "the peak of each severity stage was varied than other 431 classes". It might be better to put the full comparison "was more (or less) varied".

Figure 5 (lines 498 - 502) should be in two dimensions. The third dimension does not contribute anything, since it repeats what is indicated by the color, deforms the size and proportions of the values, and, in this case, hides the data of the series at the bottom (Quadratic Discriminant Analysis). Perhaps it would be better represented by lines rather than bars.

You should check the references as indicated in the journal's guidelines, which state "References must be numbered in order of appearance in the text (including table captions and figure legends) and listed individually at the end of the manuscript" and your references are not numbers.

In addition, there are three citations that do not appear in the list of references:
Line 536 "Abdulridha, Batuman, and Ampatzidis 2019"
Line 538 "Mewes, Franke, and Menz 2011"
Line 544 "Mewes et al. 2009"

Reviewer 2 Report

Dear authors,
Is the intention to specify different degrees of disease an intelligent way to use chemicals? Are only diseased and healthy plants sprayed differently? With different active ingredients? Is monitoring useful for early warning and preventive fungicide treatment of plants with first symptoms? What if moderately to heavily infested plants are discovered? Why do we need such an elaborate classification of the disease: class 0 (healthy, severity 0), class 1 (slightly infested, severity 1-15), class 2 (moderately infested, severity 16-34) and class 3 (heavily infested, highest severity)? Are the different classes treated differently, e.g. classes 0 and 1 preventively with immune triggers and classes 2 and 3 therapeutically with contact and/or systemic agents? Is it more a matter of assessing damage and paying compensation, or predicting harvests?
For example, is it possible to attach a fluorimeter to the drone to measure chlorophyll fluorescence, which is inversely proportional to photosynthetic efficiency, and to obtain additional information on the physiological state of the plants, e.g. whether they need to be irrigated or fertilised, etc.?
Could a system for monitoring and spraying plants with pesticides work automatically in the future? e.g. a drone flying over the fields, determining which parts of the field need to be treated and sending the data to a second drone hovering over the field and spraying only selected, e.g. diseased plants, thus reducing the amount of pesticides (costs) but also benefiting the environment? And when the task is done, the drones return to base and dock to recharge until the next mission is scheduled?

Reviewer 3 Report

Detecting plant disease severity could help growers and researchers study how the disease impacts cereal crops to make timely decisions. Advanced technology is needed to protect cereals to feed the increasing population with less chemical usage. This may lead to reduced labor usage and cost in the field. A hyperspectral camera mounted on an unmanned aerial vehicle (UAV) was utilized in this study to evaluate the severity of wheat stem rust disease in a disease trial containing 960 plots. Quadratic data analysis (QDA) and random forest classifier (RFC), decision tree classification, and support vector machine (SVM) were applied to select the wavelengths and spectral vegetation indices (SVIs). The trial plots were divided into four levels based on ground truth disease severities: class 0 (healthy, severity 0), class 1 (mildly diseased, severity 1-15), class 2 (moderately diseased, severity 16-34), and class 3 (severely diseased, highest severity observed). The RFC method achieved the highest overall classification accuracy (85%). For the spectral vegetation indices (SVIs), the highest classification rate was recorded by RFC, and the accuracy 76%. Green NDVI (GNDVI), Photochemical Reflectance Index (PRI), Red-Edge Vegetation Stress Index (RVS1), and Chlorophyll Green (Chl green) were selected from 14 SVIs. In addition, binary classification of mildly diseased vs. non-diseased was also conducted using the classifiers and achieved 88% classification accuracy. This highlighted that hyperspectral imaging was sensitive enough to discriminate between low levels of stem rust disease vs. no disease. Based on this study, it is possible to build a new inexpensive multispectral sensor to diagnose wheat stem rust disease accurately. This is an interesting research paper. There are some suggestions for revision.

1)       At the end of the abstract, it will be more intuitive and convincing to illustrate the qualitative results of a large number of experiments for verifying the superiority and effectiveness.

2)       The motivation is not clear. Please specify the importance of this paper.

3)       Please highlight the contributions/innovations of this paper.

4)       In the Section 2.1. Field experiment design, the thesis mentions that “Elevation was maintained by a level scale to make sure the camera is in the right position.”. What are the specific measures and corrective measures related to maintaining height used here?

5)       At the end of Section 2.6. Classification and feature selection methods, the thesis mentions that “Feature selection is the biggest challenge that most effective bands in order to determine the most sensitive band to detect stem rust disease. Feature selection would be determined based on which classification method can distinguish between healthy and nonhealthy plants. We used Scikit-learn, version 0.9.1 to provide a different feature selection method and, most important relative features.”. Using the advantages of Scikit learn version 0.9.1 and explaining its related operating content and processing procedures will better help readers understand it.

6)       At the end of Section 2.7. Testing wavelength, the thesis mentions that “Two parameters were utilized to explore in which wavelength, significant differences occurred in order to distinguish diseases and disease development stages: i) sensitivity value, which was calculated as the mean reflectance value of diseased leaves divided by the mean reflectance value of healthy leaves (at each wavelength); (ii) linear correlation coefficient (r), which visualizes the intensity of each spectrum band”. How are these two parameters comprehensively considered to obtain information about disease development journals.

7)       More technical details should be given.

8)       The experimental results are not convincing. More results of comparative experiments should be given.

9)       Make sure your conclusions appropriately reflect on the strengths and weaknesses of your work, how others in the field can benefit from it, and thoroughly discuss future work.

10)    Most of references are a little bit out of date. In the reference section, it is better to search and cite more latest research, which can better reflect the innovations of this paper.

Round 2

Reviewer 3 Report

All my concerns have been addressed. I recommend this paper for publication.